# Influence of nitrogen addition on soil respiration and soil properties in urban forests in Hefei city in China

Yueqin Chen, Honghua Ruan *

Co-Innovation Center for Sustainable Forestry in Southern China, Nanjing Forestry University, Nanjing, China

* crescentmoon1631@163.com

## Abstract

With the continuous development of urbanization, nitrogen deposition has affected the regeneration capacity of urban forest soil. This study focused on *Quercus acutissima* Carruth forest in a southern city Hefei in China. The effects of different nitrogen addition rates on both the soil respiration rate and the soil properties of urban forests were studied. In addition, the effects of seasonal changes on soil respiration were also analyzed. The experimental results showed that low nitrogen addition rate (5 g·m$^{-2}$·yr$^{-1}$) increased soil respiration rate by about 11% in summer, but decreased it by about 6% in winter, with a range of 5% to 20%. High nitrogen addition (15 g·m$^{-2}$·yr$^{-1}$) significantly increased soil temperature by about 12% in summer ($p < 0.05$), and significantly decreased soil temperature by about 13% in winter ($p < 0.05$), indicating the seasonal regulation effect of nitrogen addition on soil temperature. As a result, the soil respiration rate was $3.26 \pm 0.45$ µmol m$^{-2}$·s$^{-1}$ in summer and $0.51 \pm 0.07$ µmol m$^{-2}$·s$^{-1}$ in winter. The results indicated that the high nitrogen addition had a significant effect on soil respiration rate, which increased in summer and decreased in winter. Soil moisture content and temperature had significant effects on soil respiration rate, especially in the suburban test site. Nitrogen addition significantly increased soil total nitrogen and nitrate nitrogen contents ($p < 0.05$), but had no significant effect on soil pH value. Structural equation modeling analysis showed that nitrogen addition promoted soil microbial activity, releases heat, and thus increases soil temperature. Meanwhile, this method affected soil respiration rate, soil moisture, soil total nitrogen, and nitrate nitrogen content, directly or indirectly regulating soil heterotrophic respiration and autotrophic respiration. In conclusion, this study revealed the significant effects of nitrogen addition on urban forest soil respiration and its components, which provided data support and theoretical reference for understanding the dynamic changes of urban ecosystem carbon cycle, and was of great significance for guiding future urban greening and ecological protection work.

**Data availability statement:** All relevant data are within the manuscript and its Supporting Information files.

**Funding:** The author(s) received no specific funding for this work.

**Competing interests:** NO authors have competing interests.

## 1. Introduction

As socio-economic development has accelerated, the proliferation of human activities has resulted in a significant surge in global nitrogen deposition over the past century. The nitrogen input in terrestrial ecosystems has nearly quintupled, exerting a substantial influence on the global nitrogen cycle [1]. Excessive nitrogen input may lead to soil acidification and biodiversity decline [2,3].

Soil respiration (RS) is the process by which carbon dioxide ($CO_2$) is released from the soil, and RS releases about ten times as much $CO_2$ as fossil fuel emissions [4]. $CO_2$ is released mainly through the respiration of microorganisms and plant roots, which is a key link in the global carbon cycle and plays an important role in the terrestrial ecosystem [5,6]. RS is mainly affected by soil properties, soil nutrients, pH value, temperature, and moisture. The properties of soil, such as soil structure, texture, and organic matter content, directly affect the activity of soil microorganisms and roots, thereby regulating RS rate. The soil nutrients, such as nitrogen, phosphorus, and potassium, significantly affect the growth and metabolic activities of microorganisms, thereby affecting RS. Different microorganisms have different adaptation ranges to pH values. Environments that are too acidic or too alkaline can limit the activity of certain microorganisms, thereby affecting RS. Temperature and humidity are two important environmental factors affecting RS, which regulate RS rate mainly by affecting microbial activities and plant root activities [7]. Usually, an increase in temperature promotes RS. The rate of RS increases with moisture content up to a certain value near soil saturation, after which respiration decreases.

The contents of nitrogen in the soil will increase as nitrogen deposition increases, which may significantly affect the characteristics of forest soil and thus affect RS [8]. Nitrogen addition (N-addition) refers to the process in which nitrogen elements in the atmosphere settle into the soil. At present, some studies believe that N-addition can reduce the pH value of soil, thereby reducing RS [9]. However, there are also studies suggest that N-addition may increase carbon gain generated by photosynthesis of terrestrial plants, thereby increasing soil carbon content [10]. In fact, the effects of N-addition on RS are mainly realized through changes in plant root activities, microbial biomass, and soil carbon content [11]. Similar studies on the impact of urban forest RS and its composition in this article are very rare. Some researchers believe that N-addition and soil physicochemical properties have a significant impact on RS [12]. Han et al analyzed the effect of N-addition on changes in arbuscular mycorrhizal communities [13], and found that N-addition significantly reduced the overall abundance of arbuscular mycorrhizal fungi, thereby affecting the response of root biomass changes and ultimately affecting soil pH. The results indicated that N-addition indirectly reduced RS rate by reducing the abundance of arbuscular mycorrhizal fungi and altering root biomass. Hu et al. compared 100 microbial data pairs using meta-analysis methods to study the carbon utilization efficiency of microorganisms [14]. Various methods for estimating carbon were used to analyze the response information of microbial carbon utilization efficiency after N-addition. The results showed that N-addition reduced microbial carbon utilization by 12%, indicating that N-addition

affected microbial metabolism. Li et al. conducted a field experiment using temperate deciduous forests to investigate the effects of different N-addition on the dynamics of taproot in temperate forests [15]. The dynamic changes of host roots in the lower canopy and under the forest with different N-addition were compared. The results showed that higher root yield under low canopy would lead to higher RS rate [16]. Molaei et al. conducted comparative experiments based on four different soil management methods to investigate the effects of soil microbial activity and root respiration on RS in natural forests. The results showed that land use changes from natural forests to other uses had a significant impact on most research parameters.Various land use patterns can influence the proportion of particle size distribution. Both natural forests and walnut orchards exhibit elevated levels of organic matter content, and owing to their higher clay content, they possess enhanced water retention capabilities. These factors could promote soil microbial activity and root respiration, thereby increasing RS rate [17]. A total of 40 sample data were collected and their soil characteristics were measured. The results showed that poor land management led to an increase in soil pH value. Land use had a significant impact on the content of calcium carbonate in soil. Therefore, land use change had a significant impact on soil properties. Especially in China, researches on N-addition and urban forest RS are still in its early stages [18]. Currently, most researches on RS focus on agricultural systems or natural forests, while urban forests, as a special environment, are affected by human activities and high nitrogen deposition (soil total nitrogen content greater than 0.2%), and the response mechanisms of their RS and its components have not been fully studied.

In summary, research on how N-addition and physicochemical properties affect RS has been discussed by several researchers, and most of these studies are focused on agricultural systems or natural fields. However, there are significant differences between urban forests and natural forests. For example, urban forests are often affected by more human activities, higher heavy metal pollution, higher soil compaction, and different water and temperature conditions, all of which can affect RS. In addition, nitrogen deposition in urban forests is usually high, in urban forest ecosystems, nitrogen deposition is generally 25–60 kg/hm², originating from nitrogen oxides emitted from transportation, industry, and residential areas. These factors can lead to differences in nitrogen cycling in urban forest soils compared to natural forests. However, there is currently relatively little research on RS in urban forests, especially on the effects of N-addition. Therefore, in order to fill the research gap in this field, this study innovatively investigates the impact of N-addition in urban forests on RS. The study aims to reveal soil ecological processes and mechanisms that are different from natural forests, thereby enriching the theoretical system of soil ecology.

The aim of this study is to examine the impact of nitrogen addition on RS. To delve into this, the following hypotheses were formulated: 1) N-addition exerts varying effects on RS across different seasons, and 2) N-addition can influence soil pH and moisture, consequently affecting the rate of RS.

## 2. Materials and methods

### 2.1. Overview of the research site

The work was obtained the permits by Nanjing Forestry University. Nanjing Forestry University approved the field site access.

*Quercus acutissima* Carruth forest was selected as the experimental object in this study [19]. This forest was located in the suburbs of Hefei city in China. According to the distance (50 km) from the city center, the experimental area was divided into exurb experimental area and suburban experimental area. The experimental area belonged to a subtropical monsoon climate, with an average annual precipitation of about 1000 mm. The climate conditions were humid and mild, with an average relative moisture of 79%. The average annual temperature in the region was maintained at around 17°C, providing an ideal environment for the growth of *Quercus acutissima* Carruth. The exurb experimental area was located in an area at an altitude of 142 m, with a local N-addition level detection value of 16.43 kg·ha⁻¹·yr⁻¹, this meant that the amount of nitrogen applied per hectare per year was 16.43 kg. The distribution density of *Quercus acutissima* forest in this

experimental area was 837 plants per hectare, with a canopy density of 0.75. The average diameter at breast height of *Quercus acutissima* was 33.54 cm. The average tree height was 19.75 m. The suburban experimental area was located at an altitude of 92 m, with an N-addition level of 18.52 kg·ha$^{-1}$·yr$^{-1}$. The distribution density of *Quercus acutissima* forest was slightly low, with 813 plants per hectare, a canopy density of 0.70, an average diameter at breast height of 25.62 cm, and an average height of 16.48 m. The above data were obtained through on-site investigation and measurement. The average breast diameter of *Quercus acutissima* trees was measured using a tape measure at the height of the tree trunk, which was 1.3 m above the ground. The height of the *Quercus acutissima* tree was obtained using a height gauge. Density and canopy closure were calculated by counting the number of trees in the sample plot to determine the distribution density between forests. Canopy closure was evaluated through canopy coverage.

### 2.2. Research methods

**2.2.1. Experimental area selection.** The selection of the *Quercus acutissima* Carruth forest experimental area should meet the following requirements. Firstly, the area of the experimental area must be greater than 55 m × 55 m. A buffer zone of approximately 10m in width was set around the experimental area to ensure the independence of the experiment and reduce external interference. Secondly, the experimental area should have a rich variety of species, a balanced ecological environment, and a complete community structure to ensure the ecological representativeness of the experiment. In addition, the selection of experimental areas should avoid complex terrains such as trench bottoms or mountain tops. It was necessary to select areas that were relatively flat or uniformly gentle to carry out and manage experiments. Each experimental area was evenly divided into 9 plots, each of which was a 15 m × 15 m square with an area error controlled within 0.5 m. The plots were numbered sequentially from 1 to 9. Fig 1 (a) is a schematic diagram of the experimental area selection. In Fig 1 (a), starting from 2022, nitrogen deposition simulation experiments were designed and planned in the experimental area, and three different additions of nitrogen deposition were simulated on 9 plots in the experimental area.

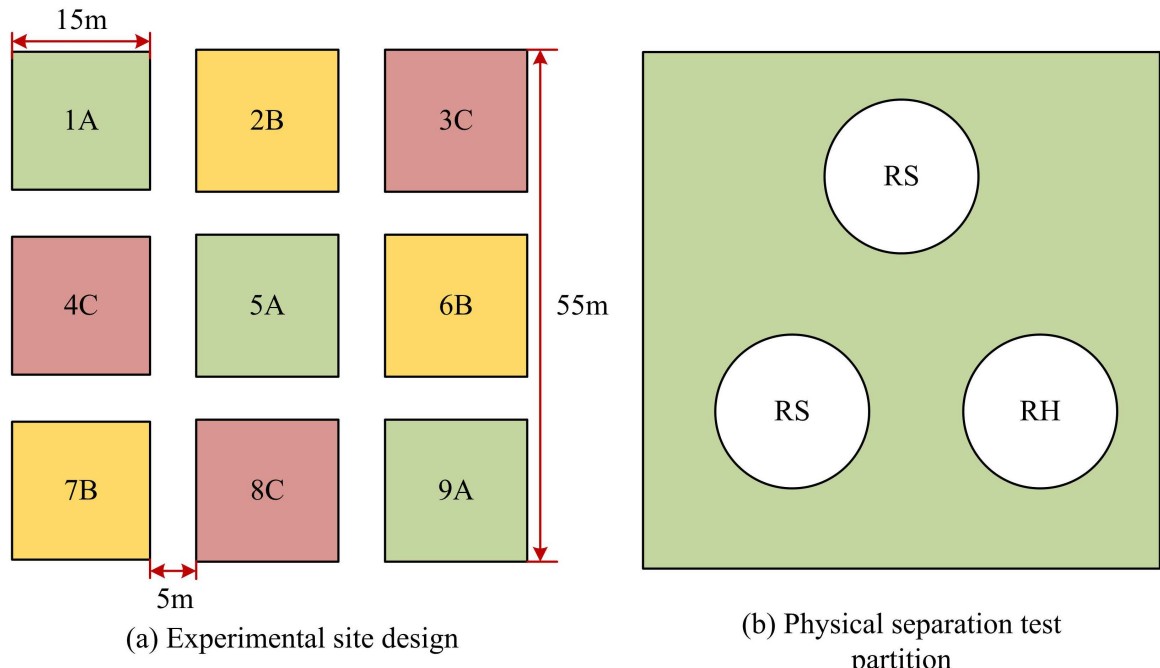

(a) Experimental site design    (b) Physical separation test partition

**Fig 1. Schematic diagram of experimental area selection.**

Zone A was the control area, and no treatment or fertilization was done during the non-growing period. Zone B was applied with low nitrogen, and the annual nitrogen application rate per square meter was 5g, that is, 5g m$^{-2}$yr$^{-1}$. High nitrogen was applied in Zone C, the annual nitrogen application rate per square meter was 15g, that is, 15 g·m$^{-2}$·yr$^{-1}$. Ammonium nitrate ($NH_4NO_3$) was a commonly-used inorganic nitrogen fertilizer that was chemically stable, easy to control and standardize, and could ensure the stability and reliability of experimental conditions when simulating nitrogen deposition. Although there might be errors in the nitrogen deposition of $NH_4NO_3$, compared with other nitrogen fertilizers, only $NH_4NO_3$ could better simulate the nitrogen deposition in natural environments, making the experimental results more practical. Therefore, the simulated nitrogen deposition fertilizer was $NH_4NO_3$. During the plant growth season, Zones B and C were fertilized every two months [20]. In addition, in order to separate heterotrophic respiration (RH) and autotrophic respiration (RA), the physical separation method was used. As shown in Fig 1 (b), the study set up three soil rings in each experimental plot, of which two rings were used to measure total RS and the other ring was used to measure RH of rootless soil by digging trenches to remove roots. When removing soil roots by digging trenches, the pH value, organic matter content, soil temperature, and humidity of the soil were first measured. After the root system was removed, the study optimized the use of excavated soil for backfilling, measured various indicators of the backfilled soil, and restored the soil to its original structure by adding organic matter or drip irrigation. The soil's bulk density remained the same as before. When restoring the original structure, the organic matter content, soil moisture, acidity, and alkalinity measured by the original structure were added with corresponding organic matter and nutrients to restore the soil to its original structure. By calculating the difference between RS and RH, the rate of RA could be obtained. The Soil Master Kit portable soil $CO_2$ rapid tester (Spectrum, USA) was used to measure the $CO_2$ content in the soil. The Latin square design was used in the experiment. By constructing a square matrix, the variation between the horizontal and in-line unit groups was separated from the experimental error. Therefore, the experimental error was smaller than the random unit group design, and the experimental accuracy was higher.

**2.2.2. Collection of experimental data.** To ensure the measurement accuracy of rootless soil, 10 days after digging trenches and removing roots, soil samples were collected using a soil drill. Three sampling points were randomly selected within the sample plot. Soil samples were taken from three soil layers of 0–20 cm, 20–40 cm, and 40–60 cm, respectively. A 2 mm sieve was used for sampling after mixing the soil from the same soil layer at the three sampling points. In this study, soil samples collected from the experimental area were quickly sealed and stored in an environment of approximately 4°C to maintain microbial activity [21,22]. When sample preprocessing, first, the soil was screened using a 70 mesh sieve to remove large particle impurities, then light impurities in the soil were blown away using wind power, and finally, magnetic impurities in the soil were removed using magnetic blocks. When removing other salts from the soil, electrodialysis was used to separate specific ions from the soil solution to prevent nitrate removal. These samples were placed in the experimental oven for thorough drying after air drying. Finally, these samples were screened for soil particles through a 70 mesh sieve. The drying temperature of the soil was between 105°C and 110°C. The drying time of the soil was 24 hours. The screened soil samples were thoroughly mixed, labeled, and encapsulated for subsequent analysis. Filtration by solution was studied to remove impurities and soluble salts from the soil in order to determine soil chemistry more accurately. It was true that some soluble nitrogen and other dissolved substances might be lost during filtration, but this loss was a small proportion of the total N-addition, and this was taken into account in the analysis. Meanwhile, chemical analysis of the filtrate was carried out by measuring the soil pH value, porosity, soil particle size, soil moisture, and bulk density, as well as the pH value of the filtrate during the filtration process to ensure the accuracy and reliability of the data. When the soil temperature dropped to 18–25 degrees Celsius, the pH value, soil moisture and nitrogen content of the pre-treated soil sample should be evaluated. Soil temperature was measured by setting up a soil ring in the experimental area and using the 109 soil temperature sensor. To ensure the accuracy and consistency of the data, soil temperature was measured every month from August 2022 to July 2023 to collect soil temperature data throughout the year. In addition, soil moisture was also measured simultaneously with RS rate to gain a more complete understanding of the influence of soil environment on RS rate.

## 2.3. RS measurements

An in-depth understanding of soil properties was particularly critical when conducting RS research. The study focused on the hemp oak forest on the outskirts of a southern city. The soil type in this region was sandy loam soil containing 5% particulate organic carbon, which had good water retention and permeability. The understory vegetation in the area consisted of herbaceous plants and shrubs, with the same level of vegetation coverage in the two selected areas. The soil texture was mainly loam or clay loam, with a relatively high proportion of silt particles, supplemented by an appropriate amount of clay and sand particles. This soil composition helped to maintain the stability of soil structure and the sustainable supply of nutrients, creating an ideal living environment for microorganisms and plant roots. According to the distance from the urban center, the study area was divided into suburban and outer suburbs, and the similarities and differences of soil characteristics between suburban and suburban are shown in Table 1.

From Table 1, although the two soil were similar texture, they showed significant differences in soil thickness, nutrient content, microbial community structure, and pH value. Specifically, surface soil (0−20 cm), as the main layer of RS, was rich in organic matter and active in biological activities. Due to the influence of more human activities (such as fertilizer application and garbage disposal), the soil nutrient content was higher and the microbial community was more complex and diverse. The exurbs remained relatively natural, and the microbial communities were dominated by saprophytic bacteria and rhizosphere microorganisms [23]. The natural difference of soil pH value was also affected by many factors such as soil parent material, climatic conditions and vegetation types. RS was measured using chamber method. Three soil rings were set up in each experimental area, with each ring spaced approximately 3 m apart [24]. The area of the soil ring was approximately 0.0318 m². The height of the soil collars was 9 cm, with 4 cm buried in the soil and the rest exposed to the surface [25]. RS on soil ring was measured using soil respiration analyzer (L-8100A, USA). Subtracting the measured soil RH from the RS value measured by the respiration meter could accurately determine the root respiration value of the soil. The difference obtained was the root respiration value of the soil [26]. The method for measuring RH was the trench method, which involved setting three measuring points approximately 2.5m from the edge of the sample plot. Each measured area was approximately 1m × 1m. A trench of about 1m was dug around the measuring point. The soil was backfilled to a level after dividing the tree roots with plastic cloth [27]. The depth of the root of trees in the soil (5−10 cm) was usually chosen as the measurement depth to better reflect the temperature state of the root active zone and thus determine the effect of temperature on RS. Since RS rate varied significantly throughout the day and was affected by factors such as temperature, soil moisture, and plant photosynthesis, different time periods of the day were selected to measure RS and RH. By measuring multiple times during the 8 a.m.-6 p.m. period, the daily variation characteristics of RS could be captured, providing more comprehensive data. Before measurement, it was necessary to remove garbage and shrubs near the experimental area to eliminate interference from other factors [28,29]. The initial time for RS measurement was set in August 2022. After that, the experiment was conducted every two months for a total of three times. Each measurement lasted about 100 minutes, from 8 a.m. to 6 p.m., with an average of 20 minutes measured every two hours. The repeated measurements at each site were three times a month. The RS rate varied with the seasons, and monthly measurements could reflect the impact of low measurement frequency on RS rate. To ensure the accuracy and consistency of the data, soil moisture and RS were measured at the same time period. In addition, soil temperature was measured monthly from

**Table 1. Comparison of soil properties between suburban and outer suburban areas.**

| Location | pH | Thickness of soil | Nutrient content | Microbiologic population; microflora | Agrotype |
|---|---|---|---|---|---|
| Suburban | 6.5~7.5 | Large thickness | Low nutrient content | Rich in biological content | Sandy soil, clay soil, loam soil |
| Outer Suburban | 3.5~5.5 | Small thickness | Organic matter is high | Water-absorbing bacteria, rhizobia | Sandy soil, clay soil, loam soil |

August 2022 to July 2023 to collect soil temperature data throughout the year. Three different soil environments were repeated to ensure the accuracy of the data.

## 2.4. Determination of soil chemical properties

The soil moisture was measured by gravimetric method. First, the fresh soil was poured into a clean aluminum box after being brought back to the laboratory, and the total weight of the fresh soil and aluminum box was weighed. Then, the soil was dried in the oven at 105°C to constant weight, and the total weight of the dry soil and the aluminum box was weighed again. By calculating the quality difference between fresh soil and dry soil, the moisture content of soil could be obtained. Flow injection apparatus and multi-N/C instrument were used to determine chemical properties such as nitrogen. The content of nitrate nitrogen and ammonium nitrogen in soil leached was measured with 0.5 mol/L $K_2SO_4$ extractant using a flow injection apparatus. The multi-N/C instrument measured soluble organic carbon in soil [30,31]. The content of available phosphorus was determined using the Olsen method. In addition, the pH value of the soil was determined using potentiometric analysis. The analysis of total soil nitrogen was determined using a element analyzer. A conductivity meter was used to measure the conductivity of soil solution. $Q_{10}$ represented the change in RS rate at a temperature difference of 10°C, obtained by measuring the temperature difference and respiration rate before and after.

## 2.5. Data analysis methods

During the data analysis phase, Excel software was utilized for entering the collected data, performing initial sorting tasks, and conducting fundamental descriptive statistical analyses. Then the data were tested for normality and homogeneity of variance, and the F value, $p$ value, mean deviation and confidence interval of the data were analyzed. SPSS software was used to analyze the correlation, regression and variance of the data. Moreover, to verify the significant effects of different N-addition treatments on RS rate and other parameters, one-way ANOVA was performed, and Least Significant Difference (LSD) was used for multiple comparisons to determine whether the differences among treatments were statistically significant ($p < 0.05$). Finally, MATLAB software was used to model and visualize the data.

## 3. Result

### 3.1. Influence of N-addition on RS in urban forests

**3.1.1. Influence of N-addition on RS rate.** Fig 2 shows how N-addition affected RS rate. From August to December, the RS rate in both the suburban and exurb experimental areas showed a decreasing trend ($p < 0.05$). From Fig 2 (a), in the suburban experimental area, compared with the RS rate at nitrogen deposition concentration A, the RS rate at nitrogen deposition concentrations B and C all increased in August. In October, there was a decline. The RS rates of nitrogen deposition concentration B and concentration C in August were $3.62 \pm 0.63$ µmol·m$^{-2}$·s$^{-1}$ and $3.50 \pm 0.31$ µmol·m$^{-2}$·s$^{-1}$. The RS rate increased by 11.04% and 7.36% compared with the RS rate of $3.26 \pm 0.45$ µmol·m$^{-2}$·s$^{-1}$ at concentration A at this time ($p < 0.05$). The RS rates of nitrogen deposition concentration B and concentration C in December were $0.48 \pm 0.06$ µmol·m$^{-2}$·s$^{-1}$ and $0.50 \pm 0.07$ µmol·m$^{-2}$·s$^{-1}$. The RS rate decreased by 5.88% and 1.96% compared with the RS rate of $0.51 \pm 0.07$ µmol·m$^{-2}$·s$^{-1}$ at concentration A at this time ($p < 0.05$). The low N-addition rate increased RS rate by about 5% to 20%, and there was a significant difference in the results ($p < 0.05$). Low N-addition rate promoted RS, while high N-addition rate inhibited RS, with changes of around 5%. The results showed significant differences ($p < 0.05$). According to Fig 2 (b), in the suburban experimental area, the changes in RS rate at nitrogen deposition concentrations A, B, and C were roughly the same as those in the suburbs. Compared with the RS rate at nitrogen deposition concentration A, the RS rates at nitrogen deposition concentrations B and C both increased in August. In October, there was a decline.

**3.1.2. Influence of N-addition on soil moisture and temperature.** Fig 3 shows how N-addition affected soil moisture content and temperature in urban forests. In Fig 3 (a), the moisture content of forest soil fluctuated between

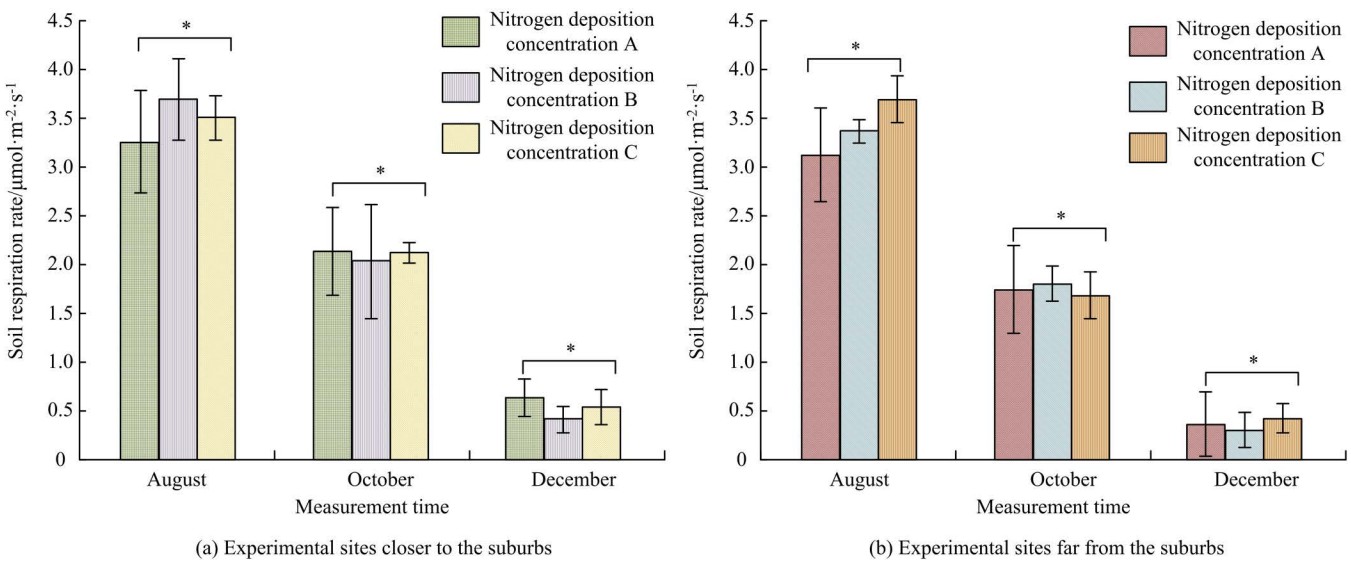

Fig 2. Influence of N-addition on RS rate(*Indicating *p* <0.05).

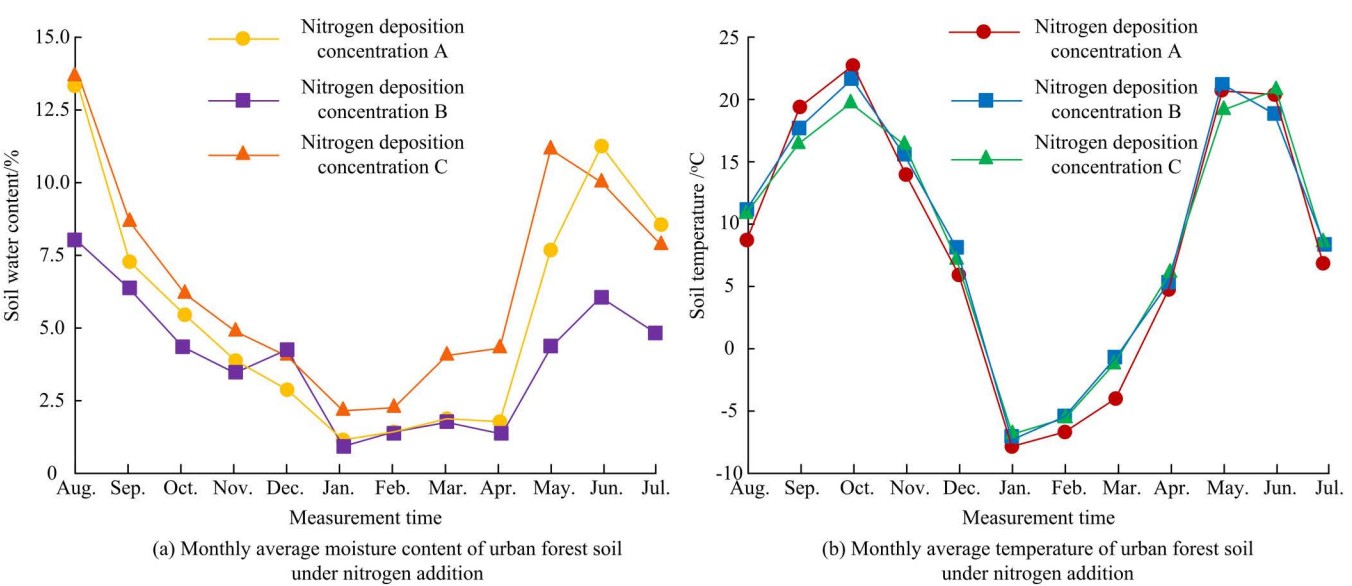

Fig 3. Influence of N-addition on soil moisture content and temperature in urban forests.

1.12% and 13.72%, reaching the lowest and highest values in January and August, respectively. In January, the soil moisture content with nitrogen deposition concentrations of A, B, and C were 1.15%, 1.12%, and 2.38%, respectively (the percentage of moisture content refers to gravimetric). In January, compared with the control group A, B decreased by only 0.03%, basically unchanged, while C increased by 1.23%. In August, the soil moisture content with nitrogen deposition concentrations of A, B, and C were 13.22%, 7.91%, and 13.63%, respectively. Compared with control group A, B decreased by 5.31% and C increased by 0.41%. Therefore, low-nitrogen reduced soil moisture content, while high-nitrogen did not affect soil moisture content obviously. The reason for this result might be that low nitrogen

treatment promoted plant growth and increased the absorption and utilization of water by plants, resulting in a relative reduction of water content in the soil. However, this reduction might not be statistically significant and was influenced by a variety of environmental factors. In Fig 3 (b), the soil temperature reached the lowest and highest values in January and October, respectively, with temperature fluctuations ranging from −8.01°C to 22.57°C. In January, the temperature of nitrogen deposition concentrations A, B, and C were −8.01°C, −7.50°C, and −7.03°C, respectively. Compared with control group A, the temperature increased by 6.36% and 12.23%, respectively. In October, the temperature of nitrogen deposition concentrations A, B, and C were 22.8°C, 21.9°C, and 19.8°C, respectively. Compared with control group A, the temperature decreased by 3.94% and 13.15%, respectively. The above results showed that low nitrogen had no significant effect on soil temperature, while high nitrogen increased and decreased soil temperature. The reason for this result might be that N-addition indirectly affected soil temperature by changing the physical properties of soil and the decomposition rate of organic matter. For example, N-addition might promote soil microbial activity and release heat, thereby increasing soil temperature to some extent. However, this effect was limited by a variety of factors, such as soil moisture and vegetation cover. In summary, low N-addition rate promoted plant growth, increased plant absorption of water, and reduced soil moisture content. High N-addition rate might promote the activity of soil microorganisms, release heat, and thus increase soil temperature.

### 3.1.3. Correlation analysis between RS rate and soil moisture.

Water content referred to the actual water content in the soil, which was usually expressed in the form of mass or volume. It focused on quantifying the absolute amount of water in the soil and reflected the actual storage of soil water. Soil moisture described the relative index of soil moisture, reflected the relative state of soil moisture, and emphasized the availability of water to plant roots or environmental processes. Table 2 presents the correlation between RS rate and soil moisture. When the nitrogen deposition concentration was 0, $R^2$ and $p$ were 0.993, and 0.042, respectively, indicating that RS rate and water content had an obvious correlation ($p < 0.05$). When the nitrogen deposition concentrations were 5 g·m$^{-2}$·yr$^{-1}$ and 15 g·m$^{-2}$·yr$^{-1}$, the $R^2$ slightly decreased compared to the nitrogen deposition concentration 0. This indicated that moisture content still affected RS ($p < 0.05$). The $R^2$ values under different nitrogen application concentrations in the experimental area far away from the suburbs were lower than those in the experimental area near the suburbs. Therefore, the correlation between RS rate and soil moisture content was relatively low in experimental areas far from the suburbs. N-addition could affect soil acidification and degradation of soil organic matter, reduce soil water use efficiency, and lead to changes in soil moisture.

### 3.1.4. Correlation analysis between RS rate and temperature.

The respiration rate of forest soil was closely related to soil temperature. The correlation between the two was evaluated to understand the impact of soil temperature on RS rate in Table 3. The $R^2$ and $Q_{10}$ of the suburban experimental area were generally higher than those of the exurb experimental area. This indicated that RS rate and soil temperature's correlation in the suburban experimental area was greater than that in the exurb experimental area. When nitrogen was added to concentration A in the suburban experimental area, the $R^2$ and $Q_{10}$ were 0.786 and 2.05, respectively. This indicated that RS rate and soil temperature had a positive correlation obviously ($p < 0.05$). When nitrogen was added at concentration B, the $R^2$ and $Q_{10}$ were 0.876

**Table 2. Correlation between the RS rate and soil moisture.**

| Experimental area | Nitrogen deposition concentration | Water content(%) | Soil moisture(%) | Parameters | |
|---|---|---|---|---|---|
| | | | | $R^2$ | *p*-value |
| Experimental areas closer to the suburbs | A | 6.8 | 28.1 | 0.993 | 0.042 |
| | B | 4.1 | 31.9 | 0.959 | 0.026 |
| | C | 7.4 | 26.6 | 0.956 | 0.031 |
| Experimental areas far from the suburbs | A | 6.9 | 14.0 | 0.632 | 0.004 |
| | B | 8.5 | 12.7 | 0.559 | 0.016 |
| | C | 5.2 | 15.7 | 0.700 | 0.044 |

**Table 3. Correlation between the RS rate and temperature.**

| Experimental area | Nitrogen deposition concentration | Inter-cept | Slope | Parameters | | |
|---|---|---|---|---|---|---|
| | | | | $R^2$ | $Q_{10}$ | *p*-value |
| Experimental areas closer to the suburbs | A | 0.223 | −1.545 | 0.786 | 2.05 | 0.039 |
| | B | 0.283 | −2.490 | 0.876 | 2.11 | 0.008 |
| | C | 0.210 | −1.510 | 0.783 | 2.00 | 0.029 |
| Experimental areas far from the suburbs | A | 0.147 | −1.373 | 0.769 | 0.97 | 0.024 |
| | B | 0.117 | −0.894 | 0.633 | 0.86 | 0.036 |
| | C | 0.178 | −1.866 | 0.883 | 0.18 | 0.049 |

and 2.11, respectively, which were 11.45% and 2.92% higher than when nitrogen was added at concentration A. The increase in nitrogen deposition concentration enhanced RS's sensitivity to soil temperature, and the enhancement of RS could affect the decomposition and release of organic carbon, leading to a decrease in organic carbon content in the soil. From the exurb experimental area, the $R^2$ was generally low, indicating a weak correlation between RS rate and soil temperature. When nitrogen was added at concentration A, although the *p*-value was 0.024, the $R^2$ and $Q_{10}$ were 0.769 and 0.97, respectively, indicating lower values.

### 3.2. Influence of N-addition on soil properties

**3.2.1. Influence of N-addition on soil pH value.** Fig 4 shows the effect of N-addition on the soil pH. From August to January, the pH values of nitrogen deposition concentrations 0, 5 g·m$^{-2}$·yr$^{-1}$, and 15 g·m$^{-2}$·yr$^{-1}$ were 8.297±0.035, 8.280±0.022, and 8.165±0.019, respectively. Compared with concentration 0, the pH of 50 g·m$^{-2}$·yr$^{-1}$, and 100 g·m$^{-2}$·yr$^{-1}$ decreased by 0.20% and 1.59%, respectively. From February to July, the pH values of nitrogen deposition concentrations 0, 5 g·m$^{-2}$·yr$^{-1}$, and 15 g·m$^{-2}$·yr$^{-1}$ were 8.283±0.015, 8.224±0.020, and 8.218±0.021, respectively. It was obvious that if there was a continuous supply of nitrogen, the content of this element in the soil increased ($p > 0.05$).

**3.2.2. Influence of N-addition on total soil nitrogen.** Fig 5 shows how N-addition affected total soil nitrogen. During a period from August to January, the total nitrogen of nitrogen deposition concentrations 0, 5 g·m$^{-2}$·yr$^{-1}$, and 15 g·m$^{-2}$·yr$^{-1}$ was 2.10±0.05 g/kg, 2.16±0.14 g/kg and 2.12±0.16 g/kg, respectively. Compared with concentration 0, the total nitrogen of 5 g·m$^{-2}$·yr$^{-1}$, and 15 g·m$^{-2}$·yr$^{-1}$ increased by 2.85% and 0.95%, respectively. From February to July, the total nitrogen of nitrogen deposition concentrations 0, 5 g·m$^{-2}$·yr$^{-1}$, and 15 g·m$^{-2}$·yr$^{-1}$ was 2.10±0.11 g/kg, 2.22±0.18 g/kg, and 2.21±0.14 g/

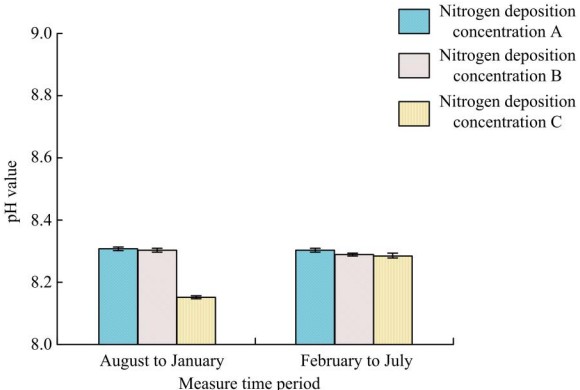

**Fig 4. Influence of N-addition on soil pH value(*Indicating *p*<0.05).**

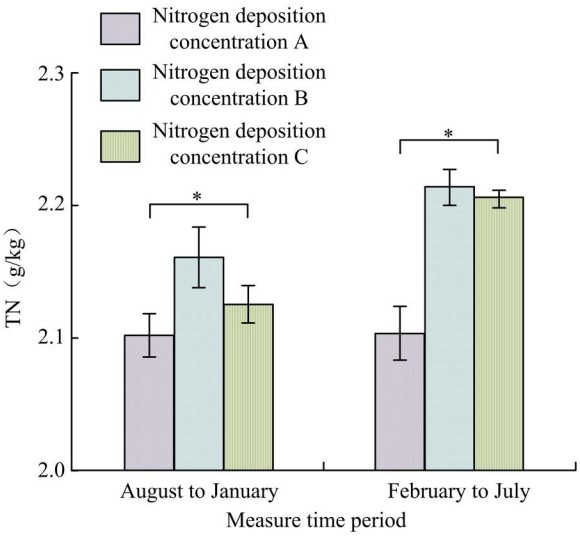

**Fig 5. Influence of N-addition on total soil nitrogen(*Indicating *P*<0.05).**

kg, respectively. Compared with concentration 0, the total nitrogen of 5 g·m$^{-2}$·yr$^{-1}$, and 15 g·m$^{-2}$·yr$^{-1}$ increased by 5.71% and 5.23%, respectively. Therefore, the total nitrogen of the soil treated with low and high-nitrogen showed a slight increase compared to the total nitrogen content of the control group soil with nitrogen blank ($p < 0.05$).

**3.2.3. Influence of N-addition on soil NO$_3$-N.** In the Fig 6, during a period from August to January, the NO$_3$-N deposition concentrations A, B, and C were 1.92±0.01 mg/kg, 3.14±0.07 mg/kg and 45.78±4.34 mg/kg, respectively. Compared with concentration A, the NO$_3$-N of B increased by 1.22 mg/kg, and the NO$_3$-N of C increased by 43.86 mg/kg. From February to July, the NO$_3$-N deposition concentrations A, B, and C were 6.98±0.07 mg/kg, 8.05±0.02 mg/kg, and 15.11±0.01 mg/kg, respectively. Compared with concentration A, the NO$_3$-N of B increased by 1.07 mg/kg, and the NO$_3$-N of C increased by 8.13 mg/kg. Therefore, the soil NO$_3$-N in the low-nitrogen and high-nitrogen treatments increased obviously compared to the nitrogen blank control group ($p < 0.05$).

**3.2.4. Influence of N-addition on soil ammonium nitrogen.** Fig 7 shows how N-addition affected soil ammonium nitrogen. From August to January, the ammonium nitrogen deposition concentrations A, B, and C were 0.61±0.12 mg/kg, 1.41±0.21 mg/kg and 1.12±0.18 mg/kg, respectively. Compared with concentration A, the ammonium nitrogen of B increased by 0.80 mg/kg, and the nitrate nitrogen of C increased by 0.51 mg/kg. From February to July, the ammonium nitrogen deposition concentrations A, B, and C were 1.05±0.15 mg/kg, 1.15±0.41 mg/kg, and 1.20±0.08 mg/kg, respectively. Compared with concentration A, the ammonium nitrogen of B increased by 0.10 mg/kg, and the nitrate nitrogen of C increased by 0.15 mg/kg. Therefore, the ammonium nitrogen in the soil treated with low and high-nitrogen showed a significant increase compared to the control group soil with nitrogen blank from August to January. There was a slight increase from February to July ($p < 0.05$).

**3.2.5. Correlation analysis of N-addition on soil properties.** Table 4 shows different N-additions and soil properties' correlation. The chemical properties of soil, including N-addition, nitrate nitrogen, and ammonium nitrogen, showed significant changes under the comprehensive effects of treatment, time, and experimental area effects ($P < 0.05$). The pH was not related to any influencing factors ($p > 0.05$). The total nitrogen content was significantly affected by time, experimental area effect, treatment and experimental area interaction, time, and experimental area interaction factors ($p < 0.05$). The $p$-value of the carbon to nitrogen ratio under the experimental area effect was 0.043. The $p$-value under the

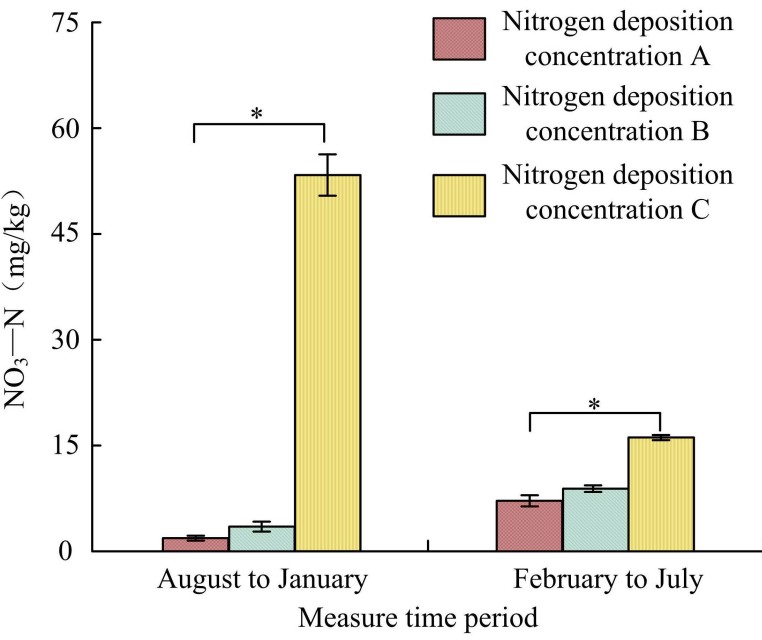

**Fig 6. Influence of N-addition on soil nitrate nitrogen(*Indicating *p*<0.05).**

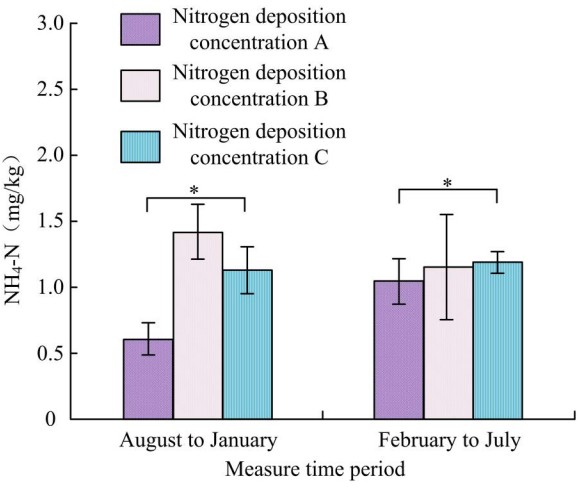

**Fig 7. Influence of N-addition on soil ammonium nitrogen(*Indicating *p*<0.05).**

interaction of time and experimental area was 0.049. Therefore, the carbon to nitrogen ratio showed significant differences under the influence of experimental area effects and the interaction between time and experimental area. This indicated that these factors significantly affected the soil carbon to nitrogen ratio ($p<0.05$). In summary, due to the influence of N-addition on soil chemical properties such as nitrate nitrogen and ammonium nitrogen under the experimental area effect, the *p*-values were 0.012 and 0.041, respectively. Therefore, the experimental area with N-addition had a significant impact on most soil chemical properties ($p<0.05$).

**Table 4. The correlation between different N-addition and soil properties.**

| Influencing factors | p-value | | | | | |
| --- | --- | --- | --- | --- | --- | --- |
| | Time effect | Experimental area effect | Process-ing: Time | Processing: Experimental area | Time: Exper-imental area | Processing: Time: Experimental area |
| pH value | 0.472 | 0.595 | 0.514 | 0.642 | 0.561 | 0.564 |
| $NO_3$-N (mg/kg) | 0.074 | 0.012 | 0.064 | 0.112 | 0.043 | 0.049 |
| $NH_4$-N (mg/kg) | 0.059 | 0.041 | 0.068 | 0.038 | 0.028 | 0.044 |
| Total Nitrogen (g/kg) | 0.044 | 0.036 | 0.128 | 0.037 | 0.041 | 0.111 |
| Soil organic carbon (g/kg) | 0.099 | 0.048 | 0.146 | 0.157 | 0.073 | 0.099 |
| Quick acting phosphorus (g/kg) | 0.070 | 0.031 | 0.062 | 0.041* | 0.051 | 0.050 |
| Carbon to nitrogen ratio | 0.056 | 0.043 | 0.065 | 0.156 | 0.049* | 0.055 |

### 3.3. Influence of N-addition on seasonal changes in RS

**3.3.1. Seasonal difference analysis of N-addition on RS rate.** In Table 5, RS rate had no significant seasonal difference under natural nitrogen deposition when the nitrogen deposition concentration was A ($p > 0.05$). RS rate had a significant seasonal difference with the increase of N-addition rate when the nitrogen deposition concentrations were B and C ($p < 0.05$).

**3.3.2. Influence of N-addition on soil RH.** An in-depth analysis was conducted to analyze the effect of different concentrations of soil N-addition on soil RH as shown in Fig 8. In August, both the suburban and exurb experimental areas showed a slight increase in RH caused by nitrogen deposition compared to the nitrogen blank A. However, the RH rates of nitrogen deposition at different concentrations were similar in October and December. In the suburban experimental area of Fig 8 (a), the RS rates of nitrogen deposition concentrations B and C were $1.28 \pm 0.12$ μmol·m$^{-2}$·s$^{-1}$ and $1.22 \pm 0.01$ μmol·m$^{-2}$·s$^{-1}$ in August. Compared with the RS rate of $1.15 \pm 0.13$ μmol·m$^{-2}$·s$^{-1}$ at concentration A at this time, they increased by 11.30% and 6.08%, respectively ($p < 0.05$). In the exurb experimental area in Fig 8 (b), the RS rates of nitrogen deposition concentration B and concentration C in August were $0.94 \pm 0.04$ μmol·m$^{-2}$·s$^{-1}$ and $1.04 \pm 0.04$ μmol·m$^{-2}$·s$^{-1}$. Compared with the RS rate of $0.88 \pm 0.21$ μmol·m$^{-2}$·s$^{-1}$ at concentration A at this time, it increased by 6.81% and 18.18%, respectively ($p < 0.05$). In summary, different N-additions had an impact on the RH rate of soil [32].

**3.3.3. Correlation analysis between RH rate with different N-additions and seasons.** The RS p-value represented the strength of the correlation between nitrogen concentration and RS. The RA p-value represented the strength of the correlation between nitrogen concentration and RA. The RH p-value indicated the strength of the correlation between nitrogen concentration and soil relative humidity. The nitrogen concentration affected RS, RA, and RH to obtain correlation. In Table 6, in the suburban experimental area, when the concentration of nitrogen deposition was A, the RS, RA, and RH were 0.621, 0.795, and 0.559, respectively, and the seasonal p-value was 0.046. This indicated a significant

**Table 5. Seasonal differences in RS rate under different N-addition.**

| Experimental area | Nitrogen deposi-tion concentration | p-value | | |
| --- | --- | --- | --- | --- |
| | | Day 1 to Day 60 | Day 60 to Day 120 | Day 120 to Day 150 |
| Experimental areas closer to the suburbs | A | 0.061 | 0.068 | 0.094 |
| | B | 0.006 | 0.001 | 0.026 |
| | C | 0.016 | 0.003 | 0.031 |
| Experimental areas far from the suburbs | A | 0.051 | 0.092 | 0.090 |
| | B | 0.040 | 0.008 | 0.022 |
| | C | 0.031 | 0.011 | 0.036 |

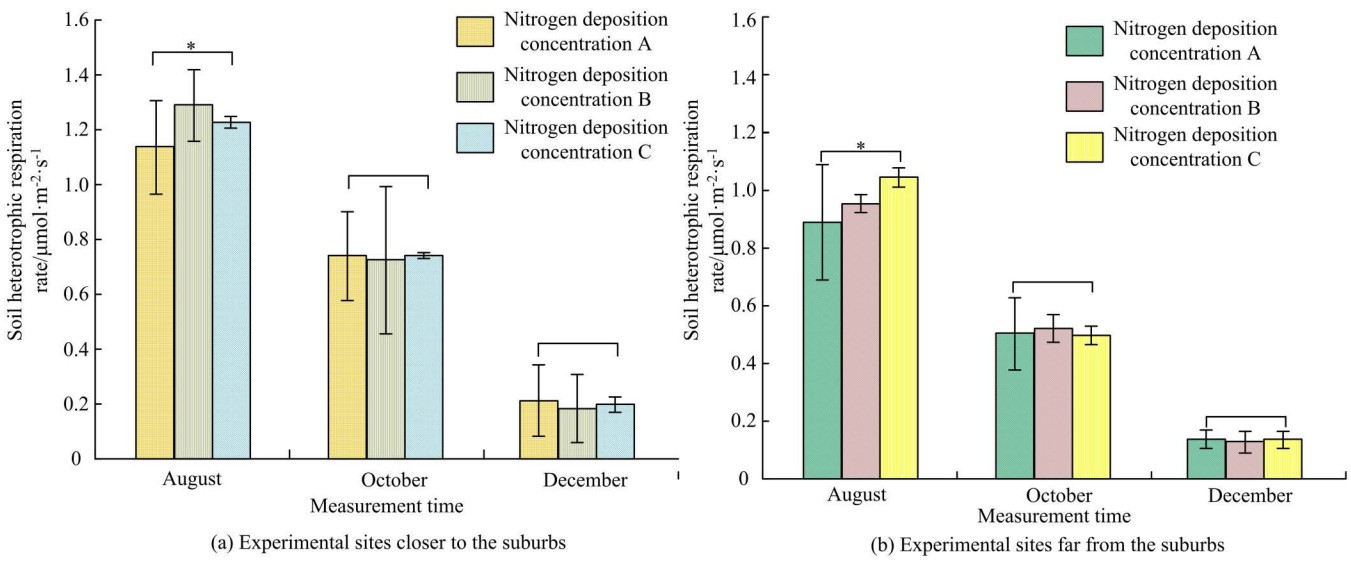

Fig 8. **Influence of N-addition on soil RH.** (* indicates *p*<0.05).

**Table 6. Correlation analysis between different N-addition and RS rates with seasons.**

| Experimental area | Nitrogen deposition concentration | *p*-value | | | |
|---|---|---|---|---|---|
| | | RS | RA | RH | Season |
| Experimental areas closer to the suburbs | A | 0.621 | 0.795 | 0.559 | 0.046 |
| | B | 0.561 | 0.625 | 0.492 | 0.032 |
| | C | 0.349 | 0.684 | 0.450 | 0.021 |
| Experimental areas far from the suburbs | A | 0.297 | 0.526 | 0.335 | 0.049 |
| | B | 0.154 | 0.449 | 0.295 | 0.030 |
| | C | 0.335 | 0.395 | 0.264 | 0.028 |

correlation between RS and seasonal changes in the region, especially a high correlation between RH and seasonal changes ($p$<0.05). In the exurb experimental area, under different nitrogen deposition concentration conditions, the seasonal $p$-value was less than 0.05. Therefore, although the values of indices such as RS, RA, and RH were relatively low, the impact of seasonal changes on RS was still obvious ($p$<0.05). Therefore, the seasonal changes of RS among different N-addition locations and fertilization cycles had obvious differences ($p$<0.05).

**3.3.4. Analysis of factors affecting RS.** In Fig 9, the red and black lines represented negative correlation and positive correlation, respectively. The solid and dashed lines indicated significant and insignificant differences, respectively. The addition of nitrogen was positively correlated with total soil nitrogen, with significant differences, and negatively correlated with soil pH, with no significant differences. The total nitrogen in soil was positively correlated with RH with significant difference. RH and RA were positively correlated with RS with significant difference. The pH of soil was significantly negatively correlated with RA. RA was also significantly negatively correlated with RH.

Various soil properties were merged into one table to observe the effects of N-addition on RS and its components in urban forests more intuitively. The correlation table between soil properties and N-addition is shown in Table 7. N-addition had a significant impact on soil properties such as temperature, moisture content, total nitrogen, nitrate nitrogen, and ammonium nitrogen, while its impact on other soil properties was not significant [33].

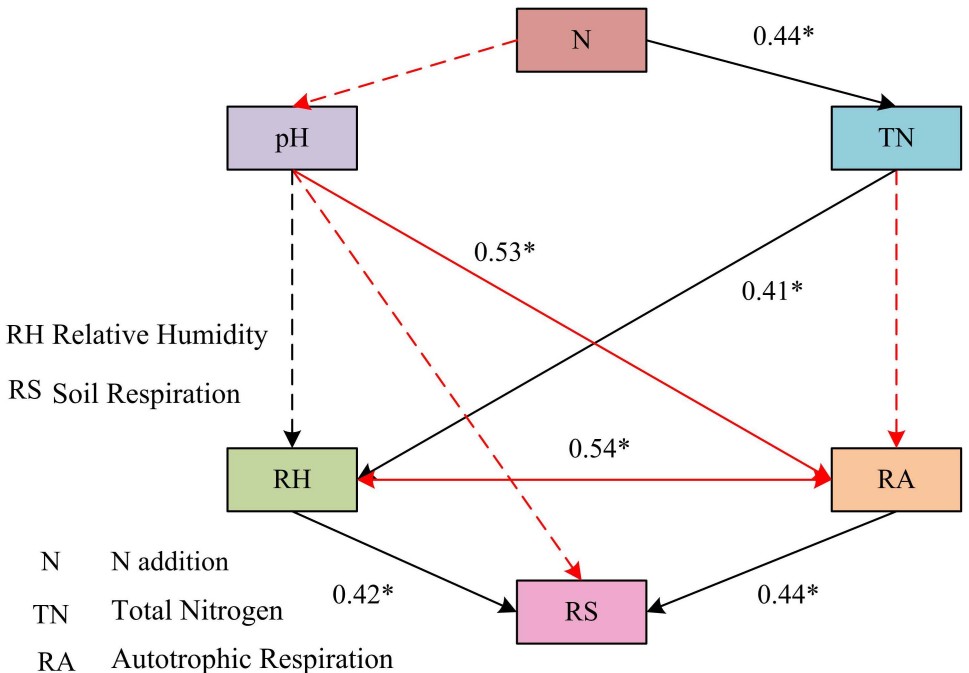

**Fig 9. A Structural equation model for analyzing factors influencing RS (\* indicates *p*<0.05).**

**Table 7. Correlation table between soil properties and N-addition.**

| Experimental area | | *p*-value of soil properties | | | | | | | |
|---|---|---|---|---|---|---|---|---|---|
| | | Soil properties | | | | RS | | Soil environmental conditions | |
| | | pH value | Total nitrogen | Nitrate nitrogen | Ammonium nitrogen | RA | RH | Moisture content | Tem-perature |
| Experimental areas closer to the suburbs | A | 0.499 | 0.049 | 0.042 | 0.011 | 0.795 | 0.559 | 0.042 | 0.039 |
| | B | 0.481 | 0.035 | 0.035 | 0.034 | 0.625 | 0.492 | 0.026 | 0.008 |
| | C | 0.464 | 0.022 | 0.003 | 0.035 | 0.684 | 0.45 | 0.031 | 0.029 |
| Experimental areas far from the suburbs | A | 0.534 | 0.044 | 0.016 | 0.029 | 0.526 | 0.335 | 0.004 | 0.024 |
| | B | 0.491 | 0.006 | 0.029 | 0.034 | 0.449 | 0.295 | 0.016 | 0.036 |
| | C | 0.381 | 0.019 | 0.031 | 0.041 | 0.395 | 0.264 | 0.044 | 0.049 |

## 4. Discussion

From the correlation analysis between RH rates with different N-addition and seasons, N-addition had a relatively low response in the short term, indicating that the effect of N-addition on RS was not significant in the short term. The reason for this finding might be that it took time for microorganisms and plant roots to adapt to increased nitrogen. The adjustment of microbial community structure, the change of enzyme activity, and the growth of plant roots all required a certain time period in response to the increase of external nitrogen, so significant changes in respiration rate might not be observed in the short term. This result was similar to Yang et al.'s study on RS and temperature sensitivity to N-addition rate, both of which indicated that the short-term effect of N-addition on RS was not significant [34]. The seasonal changes had a significant impact on RS, indicating that seasonal factors such as temperature and humidity played a more significant role

in regulating RS. Although the interaction of N-addition did have an impact on RS, the impact of N-addition was relatively small compared to the impact of seasonal changes on RS ($p<0.05$). The above results were consistent with Li et al.'s study on the seasonal changes in RS caused by water N-addition in temperate grasslands over four years [35]. This might be due to the greater fluctuation range of environmental factors brought about by seasonal changes, and the more significant interference on soil biological activities. In addition, as an important interface of soil-atmosphere carbon exchange, the growth status, species composition and physiological status of above-ground plant communities were both regulated by nitrogen supply and seasonal changes, which indirectly affected the response of RS to N-addition. The above ground plant community might play an important role in regulating the response of RS to nitrogen deposition[36]. This study found that temperature had a relatively small impact on RS. This was because N-addition could enhance soil microbial metabolism and plant root activity. The heat released by metabolism and root respiration affects soil temperature. Hu et al. used meta-analysis to compare 100 microbial data pairs and found that N-addition reduced microbial carbon utilization by 12%, indicating that N-addition affected microbial metabolism [14]. This result was consistent with the research conclusion of this article. Li et al. found that under different N-addition, the fine root yield in the lower canopy of temperate deciduous forests was twice that of understory N-addition [15]. These studies indicated that N-addition had a significant impact on RS, root biomass, microbial metabolism, and other aspects. However, the research results of this article showed that low nitrogen had no significant effect on soil temperature, while high nitrogen had a corresponding increase and decrease in soil temperature. N-addition could improve soil moisture conditions, enhance soil water persistence, promote soil water and heat functions, enhance soil permeability, and promote soil water transport [37]. This might be related to the amount of nitrogen added and the chemical changes of soil. Low nitrogen treatment might not be enough to cause significant changes in soil temperature. High nitrogen treatment might release a lot of heat through promoting microbial activity and decomposition of organic matter, thus affecting soil temperature. The above research results of this article were also similar to their conclusions. In a one-year experiment, significant differences were observed in soil organic carbon under the influence of the experimental area effect, indicating that the experimental area had a significant impact on soil organic carbon content. This was because temperature and precipitation could affect the growth and decomposition rate of plants, thus the experimental area had a significant impact on soil organic carbon ($p<0.05$). Under the experimental area effect, the $p$-value of available phosphorus was 0.031. The $p$-value under the interaction effect between processing and experimental area was 0.041. Therefore, under the influence of the experimental area effect and the interaction effect between treatment and experimental area, the difference in available phosphorus was significant ($p<0.05$). When the nitrogen deposition concentrations were B and C, although the RS, RA, and RH values were lower than the concentration A, the $p$-values were all less than 0.05. Therefore, seasonal changes still significantly affected RS under nitrogen deposition concentration ($p<0.05$). The application of nitrogen treatment in soil promoted root respiration and microbial respiration in urban forest soil. With the increase of fertilization and seasonal changes, the trend of the effect of nitrogen treatment on soil was to promote microbial respiration in urban forest soil and inhibit root respiration. Therefore, the response of RS components to N-addition was non-linear. This might be related to the diversity and complexity of the soil biome. The response of different microbial populations and plant roots to nitrogen was different, and the interaction between them might also affect the overall performance of RS. Furthermore, human-induced disruptions to soil ecosystems may contribute to greater complexity in the response of RS components to N-addition. It cannot be discounted that the impact of human activities on these components may negate their individual responses to nitrogen treatment. Therefore, it can be concluded that nitrogen was not a limiting factor for soil. Due to the balance ability of the ecosystem, N-addition enhanced the activity of microorganisms in the soil, thereby enhancing RS.

## 5. Conclusion

A nitrogen deposition simulation experiment was conducted in *Quercus acutissima* forests in the suburbs of Hefei to explore the effects of N-addition on RS and its components in urban forests. The experimental results showed that low

N-addition rate supplementation (5 g·m⁻²·yr⁻¹) significantly promoted RS in summer, increasing RS rate by about 11% ($p < 0.05$), but decreasing RS rate by about 6% ($p < 0.05$) in winter, with a range of 5% to 20%. High N-addition rate (15 g·m⁻²·yr⁻¹) significantly increased soil temperature by about 12% in summer ($p < 0.05$), and significantly decreased soil temperature by about 13% in winter ($p < 0.05$), indicating the seasonal regulation effect of nitrogen concentration on soil temperature. N-addition significantly increased soil total nitrogen and nitrate nitrogen contents ($p < 0.05$), but had no significant effect on soil pH value. Structural equation model analysis showed that N-addition directly or indirectly regulated soil RH and RA by affecting soil nutrient status. Under the condition of natural nitrogen deposition (N concentration A), the seasonal difference of RS rate was not significant ($P > 0.05$), but with the increase of N concentration to B and C, the seasonal difference became significant ($p < 0.05$). In conclusion, compared with untreated control group, nitrogen supplementation (50 g N·m⁻¹ and 100 g N·m⁻¹) significantly increased RS rate by about 11% and 7.36% in summer ($p < 0.05$), and decreased by about 6% and 1.96% in winter ($p < 0.05$), respectively. N-addition was most likely to affect RS by promoting soil microbial metabolism and plant root activities. Meanwhile, changes in soil properties caused by N-addition also indirectly affected soil temperature, which in turn affected RS. Future research should comprehensively evaluate the impact of nitrogen deposition on RS and ecosystems, in order to better understand and predict the response and adaptation mechanisms of soil ecosystems in the process of urbanization. In addition, in future studies, it is necessary to include weather parameters such as rainfall and temperature into the observation range, and in-depth analysis of the interaction mechanism between these parameters and RS and soil properties. The comprehensive assessment of the combined effects of N-addition and weather factors is expected to reveal the dynamic change law of soil ecosystem in the process of urbanization, and provide a more scientific and comprehensive theoretical basis and practical guidance for urban greening and ecological protection.

## Supporting information

**S1 File: Minimal Data Set Definition.**
(DOC)

## Author contributions

**Investigation:** Yueqin Chen, Honghua Ruan.

**Supervision:** Honghua Ruan.

**Writing – original draft:** Yueqin Chen.

**Writing – review & editing:** Honghua Ruan.

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
