## [Decision Letter · Decision Letter 0]

16 Dec 2024

Dear Dr. Ruan,

Thank you for submitting your manuscript to PLOS ONE. After careful consideration, we feel that it has merit but does not fully meet PLOS ONE’s publication criteria as it currently stands. Therefore, we invite you to submit a revised version of the manuscript that addresses the points raised during the review process.

**ACADEMIC EDITOR: ** The study is interesting while the manuscript has some problems as suggested by the reviewers. The authors should respond to the comments of the reviewers one by one and revise the manuscript accordingly. The revised manuscript would be sent to the reviewers for further reviewing.

We look forward to receiving your revised manuscript.

Kind regards,

Jian Liu

Academic Editor

PLOS ONE

Journal Requirements:

“NO authors have competing interests”

5. We note that your Data Availability Statement is currently as follows: All relevant data are within the manuscript and its Supporting Information files.

6. PLOS requires an ORCID iD for the corresponding author in Editorial Manager on papers submitted after December 6th, 2016. Please ensure that you have an ORCID iD and that it is validated in Editorial Manager. To do this, go to ‘Update my Information’ (in the upper left-hand corner of the main menu), and click on the Fetch/Validate link next to the ORCID field. This will take you to the ORCID site and allow you to create a new iD or authenticate a pre-existing iD in Editorial Manager.

7. We note that Figure 1 in your submission contain map/satellite image which may be copyrighted. All PLOS content is published under the Creative Commons Attribution License (CC BY 4.0), which means that the manuscript, images, and Supporting Information files will be freely available online, and any third party is permitted to access, download, copy, distribute, and use these materials in any way, even commercially, with proper attribution. For these reasons, we cannot publish previously copyrighted maps or satellite images created using proprietary data, such as Google software (Google Maps, Street View, and Earth). For more information, see our copyright guidelines: http://journals.plos.org/plosone/s/licenses-and-copyright.

Reviewers' comments:

Reviewer's Responses to Questions

**Comments to the Author**

1. Is the manuscript technically sound, and do the data support the conclusions?

Reviewer #1: No

Reviewer #2: Partly

2. Has the statistical analysis been performed appropriately and rigorously?

Reviewer #1: Yes

Reviewer #2: Yes

3. Have the authors made all data underlying the findings in their manuscript fully available?

Reviewer #1: No

Reviewer #2: No

4. Is the manuscript presented in an intelligible fashion and written in standard English?

Reviewer #1: No

Reviewer #2: Yes

Reviewer #1: This study was designed to study the impact of N deposition on urban forest soil respiration and soil properties. The authors selected two sites in a city in China, and measured soil respiration and its components, along with other soil properties under three N addition rates (to simulate N deposition) over one year. They reported these values and found that N additions significantly influence soil respiration but had different impacts in summer and winter. N addition influenced soil nutrient status, thus, influenced soil heterotrophic and autotrophic respiration.

I this this study provides some useful data for soil respiration in urban forests, as these data were relatively less reported compared to natural or managed forests. However, the manuscript was poorly written, with many awkward sentences. It was verbose and lengthy and needed to be thoroughly revised. I strongly suggest the authors to find a native English speaker or an editing company to edit the manuscript. The tables and figures need to be self-explanatory. Detailed captions should be provided. I recommend the authors read some published papers and see how the Results section was written. I recommend Major revision.

Specific comments:

The manuscript lacks line number and page number. It would help reviewer to review it by adding page and line numbers.

Please use past tense to describe the results from this study.

Abstract:

The major results were not clear. Adding N increased soil nutrients is not new and important. Please provide new and important results from this study.

Change nitrogen concentrations to nitrogen addition rates

The addition of high concentration nitrogen …: how about the soil respiration change under this condition?

“… and the results were significantly different”. It is not clear here. Significantly different among N additions?

The correlation coefficient R2: R2 is not correlation coefficient, but coefficient of determination.

Introduction:

At present, some studies believe that NA can promote soil respiration and thus reduce soil carbon

Content: I think the general consensus was that N addition could reduce soil respiration, mostly due to reduction in soil pH.

Han Y analyzed: Was this correctly cited? Han (****)?

“Li Xet” should be “Li X et”

Host roots?

Molaei A M. et al. … to research parameters: land use and land use efficient. Why discuss land use change here?

End of introduction: Please add objectives of this study, and hypotheses.

The average temperature is maintained: how to maintain temperature?

NA (Nitrogen addition): NA is not a common abbreviation. I suggest just using “N addition”.

Materials and methods:

Research site:

Two sites were selected. Please use a table to illustrate the similarity and differences between them.

Fig. 1: This map is not very useful. Please move to Supplemental document.

Experimental area design: (how to design an area), change design to selection?

How were 50 gN m-2yr-1 and 100 g N m-2 yr-1 determined?

“the study area was divided into outer suburbs and outer suburbs.”: ?

“The measured respiration heterotrophic (RH) of the soil is subtracted from the soil environmental measurement value to accurately determine the root respiration value of the soil”: this sentence is not correct. What’s soil environmental measurement value?

Dividing the tree roots with plastic cloth: separating the roots?

The soil moisture was measured by drying method: Drying method is not a correct term for the method. It is called the gravimetric method.

“Statistical analysis is conducted by using software such as Excel, SPSS, and MATLAB. First, the collected statistical testing data are organized, using Excel software for data entry, preliminary organization, and basic descriptive statistical analysis. Then, SPSS software is used to perform correlation analysis, regression analysis, and analysis of variance on the data. Finally, the MATLAB software is used to model and visualize the data.” This part is data analysis and should be moved to Data analysis below.

“LSD (minimum significant difference method)”: Least Significant Difference. Please used the correct term.

3. Results

Tables 1 and 2. If p value is shown, there is no need to label * or **. What are factor 1 and factor 2. Please add notes to the tables.

Fig. 10: please add notes for the abbreviations.

Table 6: I don’t understand these values. How were these correction coefficients calculated between soil properties and N addition (3 values)?

Reviewer #2: This work assesses the effect of N addition (simulating atmospheric N deposition) on soil respiration. The effect of soil properties is also studied. All this in an urban forest. This has been widely studied in non-urban forests and, therefore, the novelty of the work is precisely the location of the study area. In this sense, the work could provide more knowledge of the functioning of the forest ecosystem in these areas subjected to greater anthropic pressure.

I had to make my revision in the file, as the manuscript lacks continuous line numbering and it is impossible to refer to any part of the document without a line number. All these comments and some more are included in the file attached to the authors.

General comments:

There are many comments and doubts that should be resolved before the manuscript is published. The material and methods section should be improved, regarding the number of samples used and measurements made, a better description of the soil, some correlation analysis is not well explained; in the results section, there is no table with the measured soil values (only the p-value is used), the properties of the soils of the sampling zones are not contrasted; the discussion of the results requires a greater contrast with the works of other authors.

All these comments and some more are included in the file attached to the authors.

- There are no working hypotheses

- The doses applied to simulate N deposition are very high compared to those used in other similar work (e.g., see Mo, J., Zhang, W. E. I., Zhu, W., Gundersen, P. E. R., Fang, Y., Li, D., & Wang, H. U. I. (2008). Nitrogen addition reduces soil respiration in a mature tropical forest in southern China. Global Change Biology, 14(2), 403-412).

- I suggest a more thorough review of the bibliography, since there are references closely related to the study that have not been used, and could be useful for a better discussion of the results obtained in this work.

Specific remarks:

- see attached file

**Do you want your identity to be public for this peer review?** For information about this choice, including consent withdrawal, please see our Privacy Policy

Reviewer #1: No

Reviewer #2: No

---

## [Author Response · Author response to Decision Letter 1]

14 Feb 2025

The manuscript has been modified according to comments.

---

## [Decision Letter · Decision Letter 1]

16 Mar 2025

Dear Dr. Ruan,

Thank you for submitting your manuscript to PLOS ONE. After careful consideration, we feel that it has merit but does not fully meet PLOS ONE’s publication criteria as it currently stands. Therefore, we invite you to submit a revised version of the manuscript that addresses the points raised during the review process.

**ACADEMIC EDITOR:  ** The revised version has been improved a lot.  But the manuscript still has some little problems as suggested by the reviewer.

We look forward to receiving your revised manuscript.

Kind regards,

Jian Liu

Academic Editor

PLOS ONE

Journal Requirements:

Reviewers' comments:

Reviewer's Responses to Questions

**Comments to the Author**

Reviewer #1: (No Response)

Reviewer #2: (No Response)

2. Is the manuscript technically sound, and do the data support the conclusions?

Reviewer #1: Yes

Reviewer #2: Yes

3. Has the statistical analysis been performed appropriately and rigorously?

Reviewer #1: Yes

Reviewer #2: Yes

4. Have the authors made all data underlying the findings in their manuscript fully available?

Reviewer #1: Yes

Reviewer #2: Yes

5. Is the manuscript presented in an intelligible fashion and written in standard English?

Reviewer #1: No

Reviewer #2: Yes

Reviewer #1: The authors made efforts and addressed most of my concerns. I find that English/presentation still could be further improved. For example, both past tense and present tense are still used to describe the results in this study and in the Materials and Methods section. A few suggest changes are listed for the abstract. I recommend minor revision.

L13: change “low-concentration nitrogen” to “low nitrogen addition rate”? replace all concentration to addition?

L14: decreased it

L15: delete “and the results were significantly different (p<0.05)”?

L17: decrease significantly

L19: change “Under these conditions” to “As a result”? delete the

L20: “high concentration nitrogen” is not a current term.

L22-23. Had. Delete “the coefficient of determination. R2 is close to 1”.

L25-28: This sentence needs to be revised. There is also a logic issue here. How could N addition influence soil heterotrophic and autotrophic respirations by affecting soil respiration, as soil respiration is the total of them?

L28. Missing “.”

L164-165: NH4NO3: subscripts. May be errors?

Fig. 1: Experimental design. It seems that Latin Square design was used in this study. But it is not clear whether this design was considered for ANOVA.

Reviewer #2: There are some small issues that remain to be clarified.

- Lines 19-20: units are wrong; missing per unit of surface (µmol m-2 s-1)

- Line 23: determination R2

- Line 58 – 59: Nitrogen addition

- Line 72: Hu et al.

- Lines 96 - 99: This sentence is similar to the one in lines 110-113; it would be convenient to unify both in a single paragraph.

- Line 124: kilometers

- Line 135: 18.52 kg ·ha-1·yr-1.

- Line 165: NH4NO3

- Line 175: Is the soil brought to the same bulk density as the soil was before excavation? How is it restored to its original structure? Some explanation in this regard should be provided.

- Lines 199 – 202: Revise this sentence because neither porosity, soil particle size, soil moisture nor bulk density are chemical properties of soil; they are physical properties.

- Line 212: Check this data because it seems strange that the soil has 80% organic carbon.

- Lines 222 – 223: although the two soil were similar texture, they showed significant differences in soil thickness, nutrient content, microbial community structure, and pH value

- 288: Results

- Figure 2: y-axis legend: �mol m-2 s-1 (also in figure 8).

- Line 341: figure b y-axis: Soil temperature �C

- Line 353: Table 2. Columns 3 (Water content) and 4 (Soil moisture), what is the difference between them?

- Line 381: figure 4, y-axis: pH

- Table 4: NO3NH4 orNH4-N?

**Do you want your identity to be public for this peer review?** For information about this choice, including consent withdrawal, please see our Privacy Policy

Reviewer #1: No

Reviewer #2: No

---

## [Author Response · Author response to Decision Letter 2]

27 Mar 2025

The manuscript has been modified.

---

## [Decision Letter · Decision Letter 2]

13 Apr 2025

Dear Dr. Ruan,

Thank you for submitting your manuscript to PLOS ONE. After careful consideration, we feel that it has merit but does not fully meet PLOS ONE’s publication criteria as it currently stands. Therefore, we invite you to submit a revised version of the manuscript that addresses the points raised during the review process.

We look forward to receiving your revised manuscript.

Kind regards,

Jian Liu

Academic Editor

PLOS ONE

Journal Requirements:

Additional Editor Comments:

The revised version has been improved a lot.  But the manuscript still has some problems as suggested by the two reviewers. The authors should respond to the comments of the reviewers one by one and revise the manuscript accordingly.  

Reviewers' comments:

Reviewer's Responses to Questions

**Comments to the Author**

Reviewer #1: (No Response)

Reviewer #2: (No Response)

2. Is the manuscript technically sound, and do the data support the conclusions?

Reviewer #1: Partly

Reviewer #2: Yes

3. Has the statistical analysis been performed appropriately and rigorously?

Reviewer #1: No

Reviewer #2: Yes

4. Have the authors made all data underlying the findings in their manuscript fully available?

Reviewer #1: Yes

Reviewer #2: Yes

5. Is the manuscript presented in an intelligible fashion and written in standard English?

Reviewer #1: No

Reviewer #2: Yes

Reviewer #1: The manuscript still has many grammatic and presentation issues. Both past tense and present tense are still used to describe the methods in the Materials and Methods section and results in this study. I found that ANOVA might not be correctly conducted, as one-way ANOVA was used (L276), but Latin Square design was used in this study (L171). There are also many other issues such as units and style. I just listed a few below. The authors need to carefully check the whole manuscript and make substantial revision.

Specific comments:

L20: high nitrogen addition

L22-23: “, the coefficient of determination”: Delete this part.

L25: change “can promote” to promoted

L25-28: This sentence is too long. Please revise it.

Please check units and use them consistently. For example, meter or m, Kg /hm2 or kg hm-2.

L113: km or kilometers?

L116: mm?

L137: why 10 m not 10 meters?

L145: “Licensing Agency: Nanjing Forestry University)”: I don’t understand this part. Delete it.

L150-151: why both g per square meter per year and g m-2 yr-1 were used?

Please use past tense to describe the methods and results of this study.

L162: After removing …

L163: change are to were.

L163-171: please change to past tense.

L172: change are to were.

L172: “where the levels of each factor are systematically assigned to the experimental units,”: I don’t think this is a correct description for Latin square design.

L200: “Soil temperature was … using the Li-8100 respirator”: which sensor was used for soil temperature measurement? Li-8100 is for soil CO2 measurements, not temperature.

L228: “A respiration apparatus (Li-8100 produced by Li Cor in the United States)”: This is not a correct description for soil respiration instrument. Please check a published papers on the use of Li-8100 and company.

L273-273: this sentence was duplicated (L270-271).

L269-283: The data analysis is a mess and needs to be totally re-organized.

L284: 3. Results. Delete analysis.

I did not check the other parts. The authors need to thoroughly check the whole manuscript.

Reviewer #2: I thank the authors for their responses to my comments and doubts about some aspects of their work. After this first review, a few minor changes are listed:

- Lines 19-20: The units are not correct. Please, use the following: µmol m-2 s-1.

- Line 20: The results indicated that high addition of nitrogen had a…

- Lines 21-23: Soil moisture content and temperature had significant effects on soil respiration rate, especially in the suburban test site.

- Lines 105-106: the following hypotheses were proposed: 1) N-addition has different effects 106 on RS in different seasons, and 2) N-addition can affect soil pH a…

- Line 198: the pH value, soil moisture and nitrogen content of the pre-treated soil sample should be evaluated.

- Line 268: In this section you should include the factors to be analyzed and the levels of each factor.

- Figure 3 a) y-axis: express it as soil moisture or soil water content.

- Table 2: Add units to water content and soil moisture.

- Lines 351-354: In my opinion, this paragraph is not well understood. Please rewrite it to make clear the difference you make between water content and soil moisture.

**Do you want your identity to be public for this peer review?** For information about this choice, including consent withdrawal, please see our Privacy Policy

Reviewer #1: No

Reviewer #2: No

---

## [Author Response · Author response to Decision Letter 3]

23 Apr 2025

Please see "Reply to editors and reviewers"

---

## [Editor Report · Decision Letter 3]

30 Apr 2025

Influence of nitrogen addition on soil respiration and soil properties in urban forests in Hefei city in China

PONE-D-24-51919R3

Dear Dr. Ruan,

We’re pleased to inform you that your manuscript has been judged scientifically suitable for publication and will be formally accepted for publication once it meets all outstanding technical requirements.

Kind regards,

Jian Liu

Academic Editor

PLOS ONE

Additional Editor Comments (optional): All comments have been addressed.
---

## [Editor Report · Acceptance letter]

PONE-D-24-51919R3

PLOS ONE

Dear Dr. Ruan,

I'm pleased to inform you that your manuscript has been deemed suitable for publication in PLOS ONE. Congratulations! Your manuscript is now being handed over to our production team.

Kind regards,

on behalf of

Dr. Jian Liu

Academic Editor

PLOS ONE